# Folding–function relationship of the most common cystic fibrosis–causing CFTR conductance mutants

Marcel van Willigen[1], Annelotte M Vonk[2], Hui Ying Yeoh[1], Evelien Kruisselbrink[2], Bertrand Kleizen[1], Cornelis K van der Ent[2], Maarten R Egmond[1,3], Hugo R de Jonge[4], Ineke Braakman[1] (iD), Jeffrey M Beekman[2], Peter van der Sluijs[1] (iD)

**Cystic fibrosis is caused by mutations in the *CFTR* gene, which are subdivided into six classes. Mutants of classes III and IV reach the cell surface but have limited function. Most class-III and class-IV mutants respond well to the recently approved potentiator VX-770, which opens the channel. We here revisited function and folding of some class-IV mutants and discovered that R347P is the only one that leads to major defects in folding. By this criterion and by its functional response to corrector drug VX-809, R347P qualifies also as a class-II mutation. Other class-IV mutants folded like wild-type CFTR and responded similarly to VX-809, demonstrating how function and folding are connected. Studies on both types of defects complement each other in understanding how compounds improve mutant CFTR function. This provides an attractive unbiased approach for characterizing mode of action of novel therapeutic compounds and helps address which drugs are efficacious for each cystic fibrosis disease variant.**

## Introduction

Cystic fibrosis (CF) is caused by a mutation in CFTR, an anion channel and member of the ABC-transporter family. CFTR consists of two transmembrane-spanning domains (TMD1 and TMD2), two nucleotide-binding domains (NBD1 and NBD2) and an intrinsically disordered regulatory region (R). CFTR domains mostly fold co-translationally (Kleizen et al, 2005) and need extensive domain assembly to form a fully functional chloride channel.

To date, more than 1,800 mutations have been identified in the *CFTR* gene (http://genet.sickkids.on.ca). Most of these are not well characterized; therefore, the CFTR2 database on CF-causing variants was created (Sosnay et al, 2013). The initial version of CFTR2 contained 159 mutations, which account for 96% of all CF patients.

Mutations in CF can be divided into six different classes: (i) defective synthesis of full-length protein, (ii) defective protein processing and trafficking, leading to retention in the ER and degradation, (iii) defective channel opening (gating), (iv) reduced chloride permeability, (v) reduced synthesis of CFTR protein, or (vi) reduced cell-surface stability (Welsh & Smith, 1993; Castellani et al, 2008). This classification system is outdated because mutants can belong to multiple classes (Veit et al, 2016). An example of this is CFTR-F508del, a typical class-II mutant, which even when reaching the cell surface does not function properly and shows class-III and class-VI characteristics.

Clinical drugs targeting CFTR are divided into two types. The so-called correctors increase cell-surface expression by improving release from the ER (for instance by improving folding) and often also by increasing stability at the cell surface (Yang et al, 2003; Pedemonte et al, 2005; Van Goor et al, 2006, 2011). Compounds that improve channel function at the cell surface are called potentiators (Van Goor et al, 2009, 2014). Patients with class-III and class-IV mutations (R117H, R117C, G551D, G1244E, G1349D, G178R, G551S, S1251N, S1255P, S549N, and S549R, and several others) now have FDA approval to be treated with the potentiator VX-770. For a wide range of common patient mutations, VX-770 treatment improves CFTR function (Van Goor et al, 2014).

CFTR needs to obtain the proper conformation during the folding process to become a functional anion channel at the cell surface. Treatments with drugs or other small molecules aim to change the conformation in mutated proteins to a state where these can function at least partially. Understanding the functional and folding defects in CFTR mutants is crucial to better predict which drug (combination) provides the best precision medicine for CF treatment. CFTR function as measured by sweat-chloride concentration in CF patients correlates well with CFTR function in organoids derived from those patients (Dekkers et al, 2013, 2016; Noordhoek et al, 2016). The forskolin-induced swelling assay (FIS) is used to measure CFTR function in patient-derived organoids and is also used to interrogate individual responses to drug treatments

[1]Cellular Protein Chemistry, Department of Chemistry, Utrecht University, Utrecht, The Netherlands   [2]Department of Pediatric Pulmonology, Wilhelmina Children's Hospital, University Medical Center, Utrecht, The Netherlands   [3]Membrane Biochemistry and Biophysics, Department of Chemistry, Utrecht University, Utrecht, The Netherlands   [4]Department of Gastroenterology and Hepatology, Erasmus University Medical Center, Rotterdam, The Netherlands

Correspondence: p.vandersluijs1@uu.nl

(Dekkers et al, 2013, 2016). Most CFTR2 class-IV mutants are impaired in the formation of the pore that allows passage of anions (Mornon et al, 2015; Linsdell, 2017). R117C/H, R334W, and R347P class-IV mutants all have a lower overall chloride current but for different reasons (Sheppard et al, 1993). R117H features reduced open probability (Yu et al, 2016), whereas R347P gates like wild-type but has a conductance below 30% of wild-type levels, and R334W has either a very low conductance or diminished open probability (Sheppard et al, 1993). R117 is important for maintaining the open channel (Cui et al, 2014). The presence of many positively charged residues in the pore of CFTR suggests their involvement in chloride permeation. R334 has been implicated in interacting with anions at the mouth of the pore (Smith et al, 2001), whereas R347 forms a salt bridge that is important for maintaining an open channel (Cotten & Welsh, 1999; Cui et al, 2013).

R117C, R117H, and R334W mutant CFTR proteins are exported from the ER with efficiencies of 49, 165, and 98% of wild-type, respectively (Van Goor et al, 2014). By sharp contrast, R347P matures very inefficiently to ~15% of wild-type (Van Goor et al, 2014). All the previously described mutants, however, have been re-classified as mixed class-II/III/IV (Veit et al, 2016). To address the discrepancy between the 50% or higher maturation efficiency and the re-classifications, we revisited the functional properties and folding characteristics of the most common class-IV mutations in the original CFTR2 database. We investigated the folding–function relationship of R117C/H, R334W, and R347P using the FIS assay in organoids, intestinal current measurements (ICMs), and radio-labeling coupled to a protease-sensitivity assay of newly synthesized CFTR. We confirmed that R347P is not a "typical" class-IV mutation. Because of its defective function and a misfolded conformation, it more closely resembled a class-II mutation. Folding of the other class-IV mutants was similar, yet not identical to that of wild-type, explaining some differences between them and documenting the strength of biochemical approaches in combination with functional studies.

## Results

### R347P lacks residual function

To explore CFTR function of the class-IV mutants, we used ICM in Ussing chambers (Fig 1). We obtained rectal biopsies of heterozygous CF patients carrying one R117H, R347P, or E730X allele with the F508del mutation on the second allele and a R334W/R746X genotype. When CFTR is functioning properly, intestinal current measurements yield a positive deflection in the short-circuit current (Isc), reflecting electrogenic anion (mainly chloride) secretion. In the continuous presence of the sodium-channel blocker amiloride, elevation of intracellular cAMP by forskolin or IBMX (3-isobutyl-1-methylxanthine) triggered a large CFTR-mediated anion secretory current in rectal biopsies from healthy controls and from R117H/F508del individuals (Fig 1B and E), which was reduced in R334W/R746X (Fig 1C) and severely blunted in biopsies of R347P/F508del patients (Fig 1A). E730X/F508del biopsies even yielded a downward deflection, suggesting a net K$^+$ secretion with little to no

CFTR activity (Fig 1D). CFTR potentiators VX-770 and/or genistein, in the presence of saturating intracellular cAMP levels, did not further enhance CFTR-mediated anion secretion above the level induced by forskolin/IBMX. By contrast, the calcium-linked secretagogue carbachol further increased the Isc, the magnitude of the response proportional to the cAMP response. Carbachol acts mainly through activation of basolateral K$^+$ channels, membrane hyperpolarization, and enhancement of the electrical driving force for chloride exit through the CFTR channel (Fig 1A–E). Again, the response was severely blunted in R347P/F508del biopsies (Fig 1A).

### VX-770 does not cause swelling of R347P organoids

To confirm the effect of CFTR modulators on the functional activity of the class-IV mutants, we used the FIS assay in intestinal organoids generated from rectal biopsies. Forskolin-induced activation of CFTR causes swelling of the organoids, which we then exploit as a readout for CFTR residual function. The outcomes of ICM and FIS measurements correlate well (Dekkers et al, 2013; de Winter-de Groot et al, 2018), even though each patient-derived organoid is unique and each patient may respond differently to drug treatment (Dekkers et al, 2016; Noordhoek et al, 2016). CF patients have two mutated alleles for the CFTR protein and functional activity depends on the type of mutation on each allele. Consequently, in patient-derived material, both mutants likely are co-expressed, which formally complicates analysis of how individual mutations respond to drugs in the organoid system, analyzed in FIS and ICM assays. To correct for the contribution of the F508del allele, we subtracted half of the average swelling response obtained from eight reference F508del/F508del patients (Dekkers et al, 2016). Although EC50 values would be the preferred parameters to rate drug effects by, this assay does not allow their reliable calculation because CFTR mutants with low functionality respond so poorly to forskolin that a plateau is never reached, whereas wild-type CFTR and active mutants easily saturate the assay and plateau at ~3,200 AUC (Area Under the Curve) units because further swelling is not possible, leading to an underestimate of the EC50. When further swelling would be possible, the plateau would rise, moving the apparent EC50 to the right. The maximized swelling in the FIS assay, therefore, precludes proper readout of the EC50.

R347P/F508del was the only class-IV heterozygote that failed to give an appreciable functional response without modulators (Fig 2A). Importantly, preincubation of the organoids with VX-809 rescued R347P/F508del functionally, suggesting that it needs a corrector to either reach the cell surface or function at this location. Acute addition of VX-770 had similar effects as VX-809, but the strongest response of R347P/F508del was seen with both treatments (Fig 2A). Correction for the F508del allele (Fig 2A') clearly showed that the effect of VX-770 on the function of R347P is minimal and similar to that obtained with the vehicle control. This showed that the R347P allele can be rescued by VX-809 but not by VX-770 even when combined with VX-809.

Other class-IV mutants (R117H and R334W) showed CFTR function under basal conditions, which was increased by VX-770, whether corrected for the F508del allele or not (Fig 2B, B', C, C'). The R117H/F508del curve in the presence of modulators nears that of organoids from healthy controls (Dekkers et al, 2016). The apparent negative

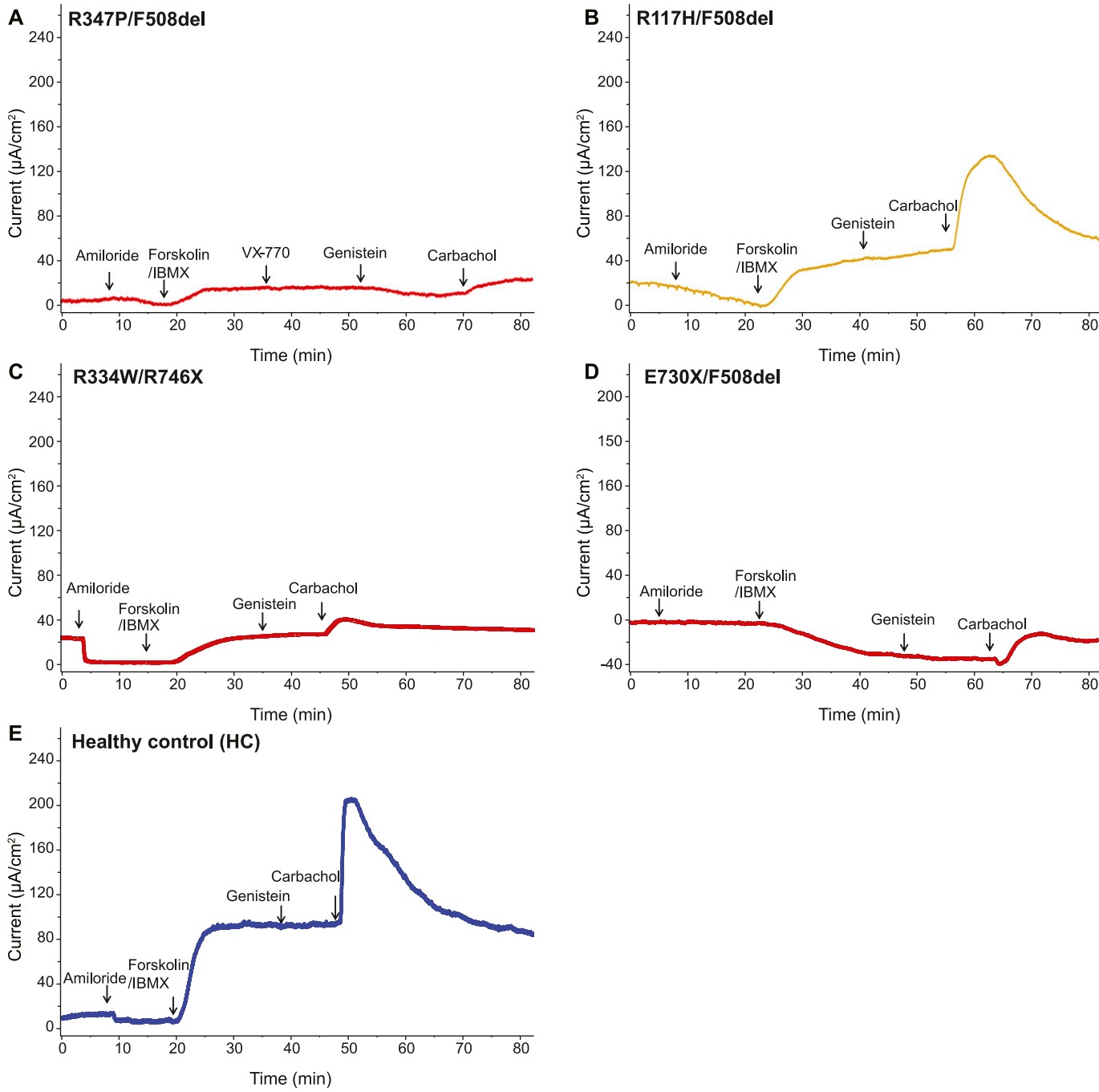

**Figure 1. R347P is inactive.**
ICM measurements of human rectal biopsies of tissues, which were treated with indicated compounds. **(A)** Tracing of an R347P/F508del patient showing no response upon forskolin/IBMX addition. Even in the presence of potentiator VX-770, there was no detectable increase in signal. **(B)** Tracing of R117H/F508del showing a response after addition of forskolin/IBMX. **(C)** Measurements of a R334W/R746X showing a slight upward deflection after forskolin/IBMX addition. **(D)** E730X/F508del patient showing a downward deflection upon forskolin/IBMX addition. **(E)** Tracing of a healthy individual, showing a large anion secretory current after forskolin/IBMX addition. Arrows indicate the addition of compounds: amiloride (100 $\mu$M), forskolin (10 $\mu$M), IBMX (100 $\mu$M), VX-770 (20 $\mu$M), genistein (50 $\mu$M), and carbachol (100 $\mu$M).

effect of VX-809 may be due to the subtraction procedure over-estimating the contribution of the F508del allele in this specific patient sample. The F508del/E730X organoid was relatively inert to VX-809 and VX-770 (Fig 2D), suggesting that the F508del allele in this patient gained little function even upon combination of modulators.

Patients with the same genotype, however, can differ in disease phenotype and this specific patient could have low residual function (Dekkers et al, 2016; Pranke et al, 2017). We concluded that the class-IV mutations on one allele contributed significantly to the swelling of organoids derived from these patients.

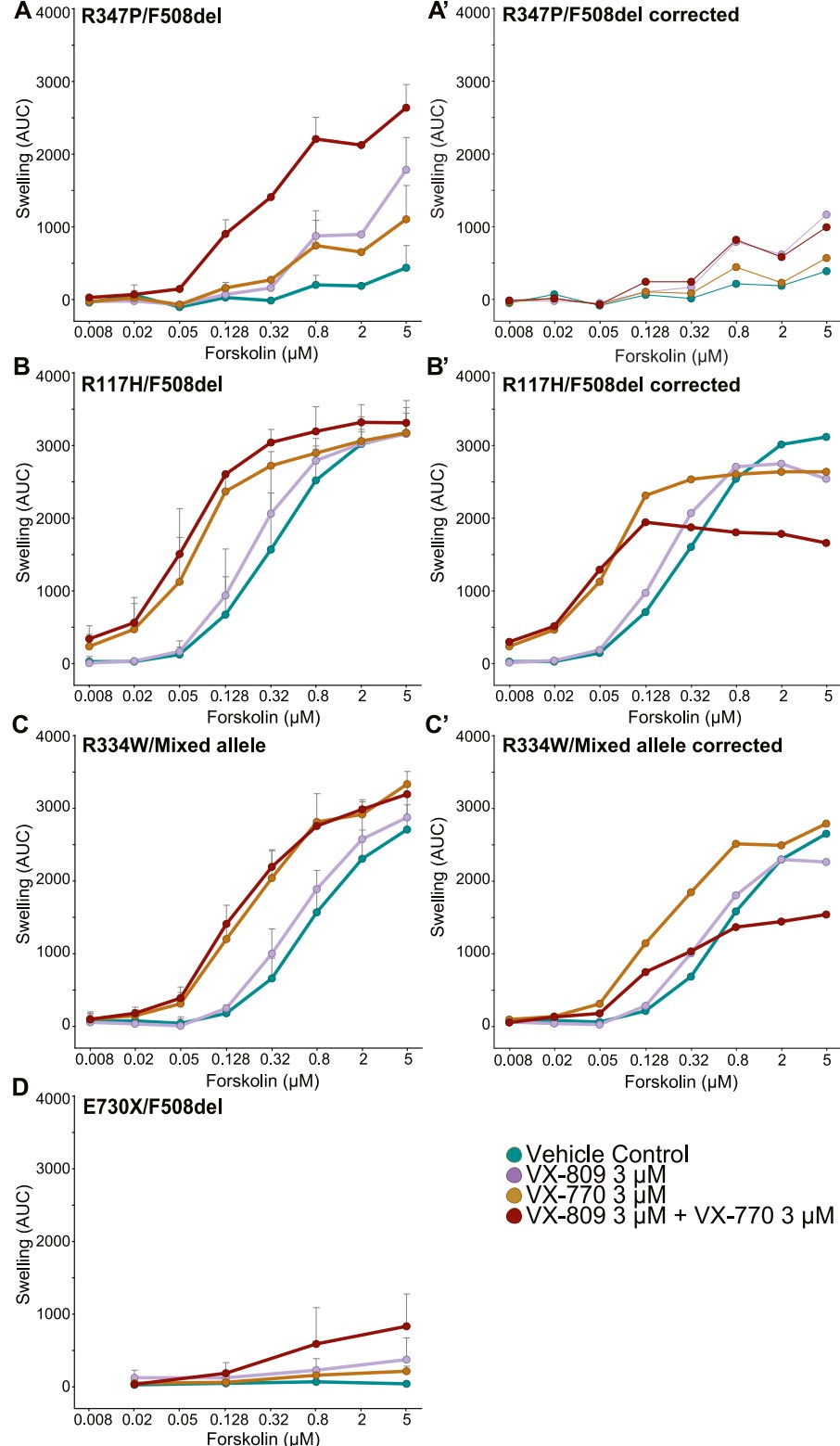

**Figure 2. R347P is functionally unresponsive to VX-770.**
Organoids of different patients were treated with an increasing concentration of forskolin, and swelling was measured after 60 min. **(A)** Nontreated R347P/F508del organoids showed a low amount of swelling, indicative for reduced CFTR function. VX-809 and VX-770 gave a moderate response, and a combination of VX-809 and VX-770 had an additive effect. **(A′)** Corrected response of A in which 50% of the average swelling value of F508del/F508del organoids from eight patients from Dekkers et al (2016) was subtracted from the values of R347P/F508del. **(B)** The R117H/F508del showed residual function. VX-770 alone or a combination of VX-770 and VX-809 gave a maximal effect, where a plateau was reached at lower concentrations of forskolin. **(B′)** R117H/F508del swelling, corrected as in A′ **(C)** R334W/Mixed allele variants with F508del, R746X, and N1303K on the second allele, had a lower but overall similar response as R117H/F508del in B. **(C′)** Same as (A′) and (B′) but subtracted from the R334W/Mixed allele. **(D)** The F508del/E730X had no residual response and a slight increase upon addition of either drug alone, which was additive to the combination of VX-809 and VX-770. **(A–C′)** Data represent average ± SD of triplicate measurements of organoids from three distinct patients. Data in (D) represent average ± SD of triplicate measurements of an organoid from one patient.

### Transport of R347P to the Golgi complex is impaired

Unlike the other tested class-IV mutants, R347P did not show appreciable residual function in the ICM and FIS assays. We, therefore, compared folding and ER-exit characteristics of R117C, R117H, R334W, and R347P in HEK293 cells. The position of the mutations is shown in a 3D representation of the human apo-CFTR structure (Liu et al, 2017) (Fig 3A). The cells were labeled with [35]S-methionine/cysteine during a 15-min pulse and subsequently chased in the presence of excess unlabeled amino acids. CFTR was immunoprecipitated directly from detergent cell lysates or after subjecting the lysates to limited proteolysis as detailed before (Kleizen et al, 2005; Hoelen et al, 2010).

Synthesis of CFTR takes on average 10 min, and at least another ~30 min is needed to fold the protein into a state where it can leave the ER (Hoelen et al, 2010; Kleizen et al, 2005). Wild-type and mutant CFTR molecules were synthesized to similar levels during the 15-min pulse (Fig 3B, top left panel) and are in the ER, where wild-type folds into an ER-export-competent state. During the chase period, the folded wild-type protein (Fig 3B, top right panel) is transported to the Golgi complex where its N-linked glycans are modified into the complex-glycosylated form typical for CFTR molecules that

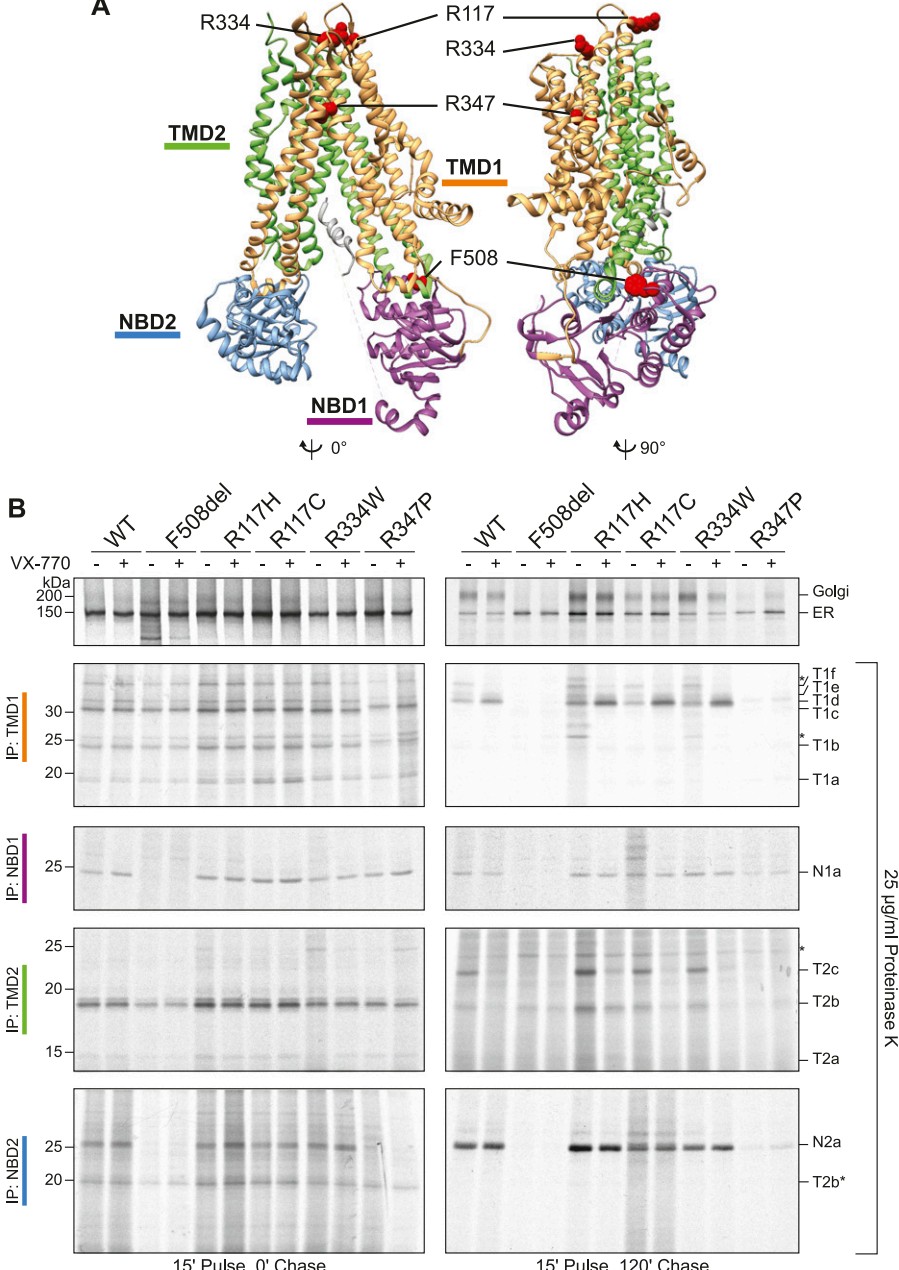

**Figure 3. VX-770 does not improve R347P folding.**
**(A)** Cryo-EM structure of human CFTR (Liu et al, 2017) (PBD: 5UAK), generated using Chimera software (Pettersen et al, 2004), illustrating that R117 is in the cytoplasm in the first extracellular loop between TM1 and TM2, R334 is located in the second extracellular loop between TM3 and TM4, and R347 is located in TM6 inside the pore. The colors wheat, purple, gray, green, and blue indicate TMD1, NBD1, R region, TMD2, and NBD2, respectively. **(B)** CFTR was expressed in HEK293 cells, which were labeled with [35]S-methionine/cysteine for 15 min and chased for 0 and 2 h in the presence or absence of 3 $\mu$M VX-770. The cells were lysed in 1% Triton X-100 in MNT, and the cell lysates were treated or not with Proteinase K at 25 $\mu$g/ml for 15 min. CFTR and fragments were immunoprecipitated using TMD1C (TMD1), Mr. Pink (NBD1 and full-length CFTR), TMD2C (TMD2), and 596 (NBD2) antibodies. * indicates nonspecific bands.
Source data are available online for this figure.

have left the ER (Cheng et al, 1990; Hoelen et al, 2010). The fraction of class-IV mutants that reached the Golgi complex was mutation specific, where R374P left the ER least efficiently. For comparison, we included F508del CFTR in the analysis (Fig 3B) as it fails to fold and reach the Golgi complex but instead is retained in the ER and disposed off through ER-associated degradation (Cheng et al, 1990; Jensen et al, 1995; Ward et al, 1995; Meacham et al, 2001).

### R347P is severely misfolded

We next compared the folding of the class-IV mutants with wild-type and F508del CFTR, by subjecting detergent lysates prepared from radioactively labeled cells to limited proteolysis followed by immunoprecipitation with CFTR domain–specific antibodies. Proteases will cleave a misfolded protein or a folding intermediate more efficiently than a well-folded protein, leading to different digestion patterns on an SDS-PAA gel (Hoelen et al, 2010; Kleizen et al, 2005; Sharma et al, 2001; Zhang et al, 1998). Immediately after the pulse, the TMD1 antibody immunoprecipitated three proteolytic fragments, identical for wild-type and mutant CFTR (Fig 3B, IP:TMD1, T1a-c). The wild-type CFTR TMD1 fragments disappeared over time during the chase, concomitant with the appearance of three larger (more protease-resistant, more folded) TMD1 fragments, T1d-f, which were absent from the Class-II F508del mutant. Digestion of the class-IV mutants gave the same three wild-type TMD1 fragments after the chase, except R347P, which only yielded a detectable bottom fragment T1d because T1e and T1f were below (Fig 3B) or close to (Figs 4 and 5) detection level. The NBD1 domain was protease resistant in both pulse and chase samples of wild-type CFTR and all four class-IV mutants (Fig 3B, IP: NBD1), with fragment quantities relating to full-length proteins as in wild-type. As expected, F508del NBD1 was protease sensitive (Hoelen et al, 2010). Immunoprecipitation with the TMD2 antibody showed a prominent fragment (T2b) immediately after the pulse, whose intensity decreased during 2 h of chase. In wild-type CFTR and again three of the four class-IV mutants, TMD2 became more protease resistant and more folded during the chase, whereas this T2c fragment was barely detectable in the R347P and even less in the F508del CFTR digests (Fig 3B, IP: TMD2, and Figs 4 and 5). The NBD2 protease-resistant fragment, N2a, indicative of NBD2 folding, became more intense during the chase period, in wild-type CFTR and the class-IV mutants, but again to a much lesser extent in R347P. In the archetypical class-II F508del mutant, NBD2 remained completely protease sensitive, that is, not folded (Fig 3B, IP: NBD2).

Whereas folding of three of the class-IV mutants was similar to that of wild-type CFTR, R347P formed a notable exception because all its domains except NBD1 were more Proteinase K susceptible, documenting that its domains were less well folded than in wild-type CFTR and the other mutants (Fig 3B). Because limited proteolysis of R347P yielded some TMD1, NBD1, TMD2, and NBD2 fragments that were completely absent in F508del (Fig 3B), we concluded that R347P was slightly better folded than F508del. Protease susceptibility of R347P's TMD1 was different in two ways: not only less of each fragment but also slightly larger fragments were generated than with wild-type CFTR. All fragments except T1a had a slightly lower mobility on SDS-PA gels (Figs 3B, 4, and 5,

TMD1b-f) because the amino-acid change affected the mobility of each R347P-containing fragment (our unpublished observations).

### Most class-IV mutants respond to VX-770 conformationally

We then assessed the biochemical effects of VX-770 and VX-809 on the class-IV mutants and established whether improvement of abnormal folding could explain the outcomes of the functional experiments. Addition of VX-770 did not affect the synthesis of wild-type CFTR or any of the mutants (Fig 3B, left panel). Whereas it slightly decreased the transport to the Golgi of some mutants (Fig 3B, right panel), steady-state CFTR levels were not affected by 3 $\mu$M VX-770 (Fig 4B). Limited proteolysis of the 2-h chase samples followed by immunoprecipitation with the TMD1 antibody showed, however, that VX-770 slightly destabilized the N terminus of TMD1 of wild-type and the R117 and R334 mutants, as the three late fragments, T1d-f, collapsed into the smallest T1d band (Fig 3B); these fragments only differ at their N terminus (our unpublished observations). In apparent contrast, the approximately twofold increase in signal of the combined T1d-f large fragments in the presence of VX-770 (Fig 3B) confirmed that VX-770 increased overall TMD1 protease resistance (Byrnes et al, 2018). Because folding of R117H, R117C, and R334W were wild-type–like in control and VX-770–treated samples, we concluded that a gross conformational defect did not account for their reduced chloride transport.

### The folding defect of R347P is rescued by VX-809

Because VX-809 increased the functional response of R347P and VX-770 had little if any effect (Fig 2A), we investigated whether VX-809 rescued folding of this mutant. The general effect of VX-809 was an increase in quantity of CFTR, of both full-length and protease-resistant fragments (Fig 4A–C; B. Kleizen, I. Braakman, unpublished observations; Van Goor et al, 2011; Okiyoneda et al, 2013; Ren et al, 2013; Loo & Clarke, 2017). This strongly suggests that VX-809 helps CFTR fold into a more protease-resistant, stable conformation. VX-809 partially rescued F508del from the ER as seen by the appearance of a Golgi form after a chase as well as the late fragments of TMD1, TMD2, and NBD2 mentioned previously (Fig 4A, right panels). VX-809 did not rescue folding of F508del NBD1 as seen in the immunoprecipitations for NBD1 (Fig 4A; Okiyoneda et al, 2013). R347P was the only mutant that responded well functionally to VX-809 in the FIS assay (Fig 2A and A'), which correlates with the increased levels of R347P after 15 min of pulse labeling, the improved transport to the Golgi complex (Fig 4A, top panels), and the increased steady-state levels (Fig 4B).

In nontreated control conditions, limited proteolysis of the R347P mutant yielded minute quantities of late TMD1 fragments. The strong improvement of R347P folding and conformation by VX-809 is demonstrated by the recovery of the late TMD1 fragments, albeit enriched for T1d (Fig 4A). The other domains also gained stability after VX-809 treatment, which hence improved the complete protein (Fig 4A) and its function (Fig 2A and A'). That VX-809 rescued the R347P mutant more efficiently than F508del was likely caused by its improving TMD1 folding, the domain that contains the R347P mutation and hence the primary defect.

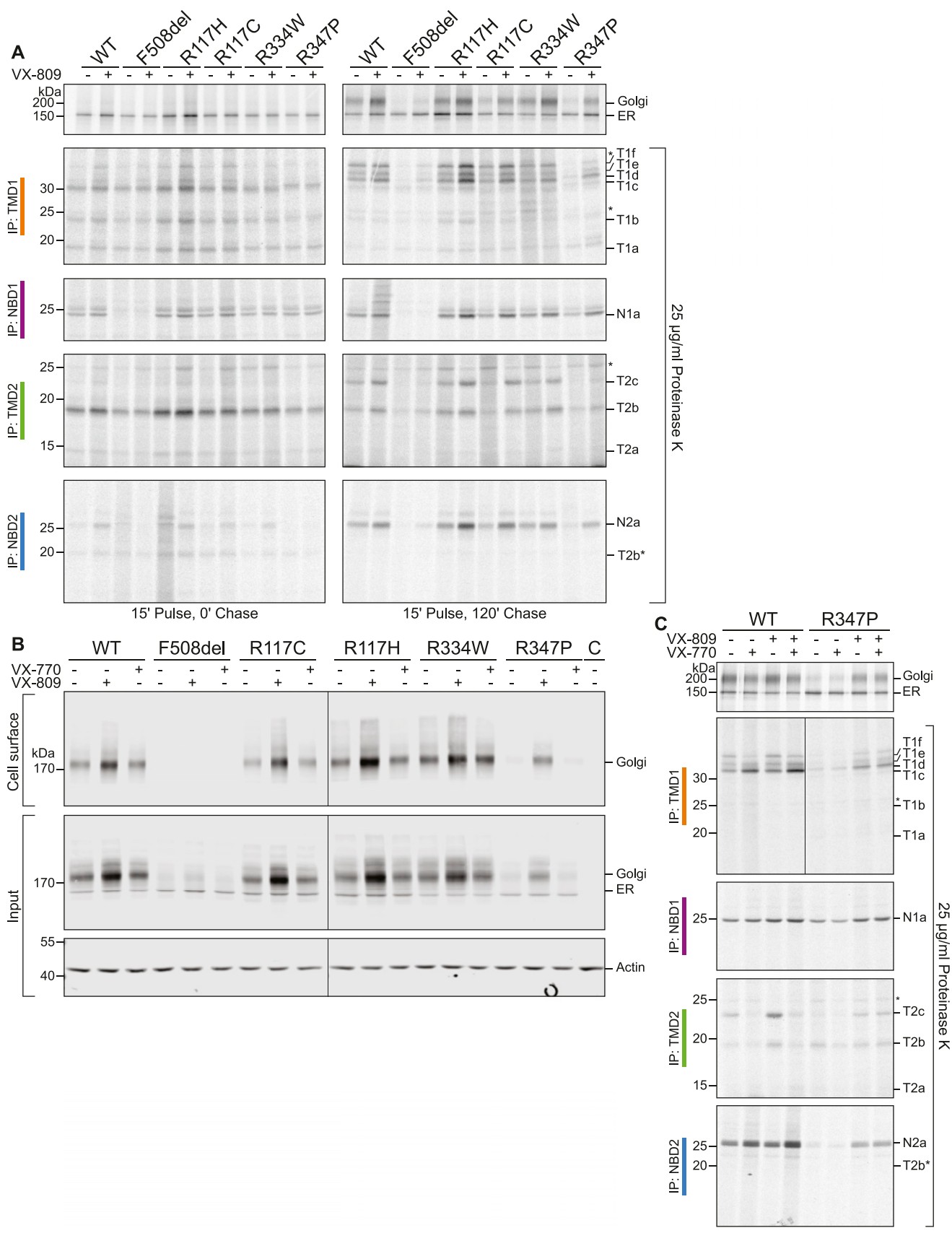

To assess whether enhanced transport of CFTR variants to the Golgi complex resulted in more molecules reaching the plasma membrane, we used cell-surface biotinylation. As shown in Fig 4B, wild-type, R117C/H, and R334W CFTRs have similar cell-surface expression, whereas R347P and F508del CFTRs were barely or not detectable. Plasma membrane expression was increased to a variable extent upon VX-809 treatment (minimally for F508del), whereas VX-770 was without effect (Fig 4B). We next investigated whether R347P CFTR that had been rescued by VX-809 became responsive to VX-770 (Fig 4C). In agreement with the functional results in the FIS and ICM assays, R347P remained unresponsive to VX-770 even though VX-809 rescued the late fragments from all domains of R347P (Fig 4C).

### The proline residue in R347P destabilizes TM6

The reduced stability of R347P is likely due to changes in transmembrane segment 6 (TM6) of TMD1, which contains the mutated residue. The question is whether the loss of the charged amino acid or gain of the proline caused the defect. To establish the role of the proline, we used another disease-causing mutant in which R347 was changed to histidine and designed three additional mutants (Fig 5A and B). A helical wheel representation showed that R347 is embedded in a polypeptide stretch enriched for small hydrophobic residues (Fig 5C). We, therefore, changed R347 to a valine or leucine, whereas the arginine-to-glutamine was made because glutamine is present at the 347th position in other ABC transporters.

The protease digestion pattern of the four mutants was similar to that of wild-type CFTR (Fig 5A). Like in R347P, mutation to a glutamine resulted in the upward shift of the T1b-f bands (Fig 5A) because of a small change in mobility of the mutation-containing fragments. The loss of arginine caused a small alteration in late TMD1 fragments of all R347 mutants, with an enrichment of T1d over T1e and T1f compared with wild-type TMD1 (Fig 5A and B). Nevertheless, relative to this minor change, the introduction of a proline residue caused the major folding defects.

## Discussion

Here, we investigated the structure–function relationship of the most common class-IV CF-causing CFTR mutants, defining a subgroup of CF patients also known as the altered-chloride-conductance group (Welsh & Smith, 1993; Castellani et al, 2008). CFTR mutants of this class can escape the ER and reach the plasma membrane. Patients with these mutations may benefit from treatment with a single potentiator drug to 'activate' the mutant CFTR protein at the cell surface. Patients carrying class-IV mutations, with the exception of R117H and R117C, do not yet have clinical approval for treatment with

the VX-770 potentiator compounds. We addressed the folding characteristics of class-IV CFTR mutants to uncover the underlying mechanism for functional effects of VX-770 and corrector VX-809 treatment in this group and found that all responded and were improved conformationally by VX-809. The corrector improved folding in all mutants but barely affected R334W and R117H function, most likely because their cell-surface levels already are at a maximum, wild-type–like level. By contrast, R347P deviated in that it benefitted well from VX-809 (in a manner distinct from F508del CFTR), yet did not have a functional VX-770 response. We have shown that all functional mutants respond to the corrector drug and that R347P responded better to corrector than to potentiator treatment, classifying R347P as bona fide class-II mutant.

An explanation of the differences between the mutants is greatly facilitated by the recent structure models of CFTR and the cryo-EM structures of CFTR and other ABC transporters, which enable structure-based experimental design and analysis. The "hyper-response" of R347P to correction is likely caused by VX-809, correcting precisely where the defect is, in TMD1 (Loo et al, 2013; Loo & Clarke, 2017; our unpublished results). R347P resides in transmembrane helix 6 and the cryo-EM structure of human CFTR shows that TM6 interacts with the N terminus at the elbow helix and packs closely to TM3 and ICL1, the first intracellular loop in TMD1, which interacts through its coupling helix with NBD1 in folded CFTR (Fig 5D) (Liu et al, 2017). Changing arginine 347 to a proline in TM6 will destabilize the helix and will likely affect the packing of TM6 with ICL1 and the N terminus (Choi et al, 2005). VX-809 was reported to promote interactions of NBD1 with ICL1 (Loo et al, 2013; Loo & Clarke, 2017), and allosterically with ICL4 (Hudson et al, 2017; Laselva et al, 2018), both consistent with our data (Fig 4A–C) and our unpublished results (Kleizen & Braakman, in preparation). Corrector VX-809 improves packing of TMD1, and thereby rescues interactions with the other domains, in particular NBD1.

The destabilization of the N terminus of TMD1 was not unique to R347P, as also VX-770 had this effect in our assays. Indeed, the N terminus and ICL1 are affected by VX-770 to some extent (Byrnes et al, 2018). We conclude that the amino terminus in R347P is destabilized to such an extent that VX-770 failed to exert a significant effect in the two functional assays. Functional experiments suggest that (heterozygous) R347P patients can benefit from a combination treatment of the two drugs, which was shown to be beneficial to F508del homozygous patients with variable improvements between 4.3 and 6.7% (Wainwright et al, 2015).

Even though the introduction of the proline residue caused the strongest defect, the highly conserved arginine does play a key role in maintaining the open pore architecture. R347 makes a salt bridge with aspartate residues in positions 924 and 993 of TMD2 in the open channel (Cui et al, 2013). Removal of this arginine, therefore, causes reduced function, as in the R347H patient mutant (Cotten & Welsh, 1999;

**Figure 4. VX-809 corrects R347P folding.**
**(A)** HEK293 cells expressing CFTR constructs were labeled for 15 min and chased for 0 or 2 h in the presence or absence of VX-809. The cells were lysed in 1% Triton X-100 in MNT, and the cell lysates were incubated with or without Proteinase K at 25 µg/ml for 15 min. CFTR and fragments were immunoprecipitated using domain-specific antibodies as in Fig 3. **(B)** Cell-surface biotinylation of CFTR in HEK293 cells with or without VX-809 or VX-770 pretreatment. The cells were lysed in 1% Triton X-100 in MNT, and the lysates were used for pull-down of biotinylated proteins with NeutrAvidin beads. 7.5% SDS-PAA gels were run and transferred to PVDF membrane and blotted against CFTR (596) or Actin. **(C)** Same as in (A) but in the presence of VX-770, VX-809, or both as indicated. All lanes of IP:TMD1 were present on one gel, but the solid black line indicates where the lanes were removed. * indicates nonspecific bands.
Source data are available online for this figure.

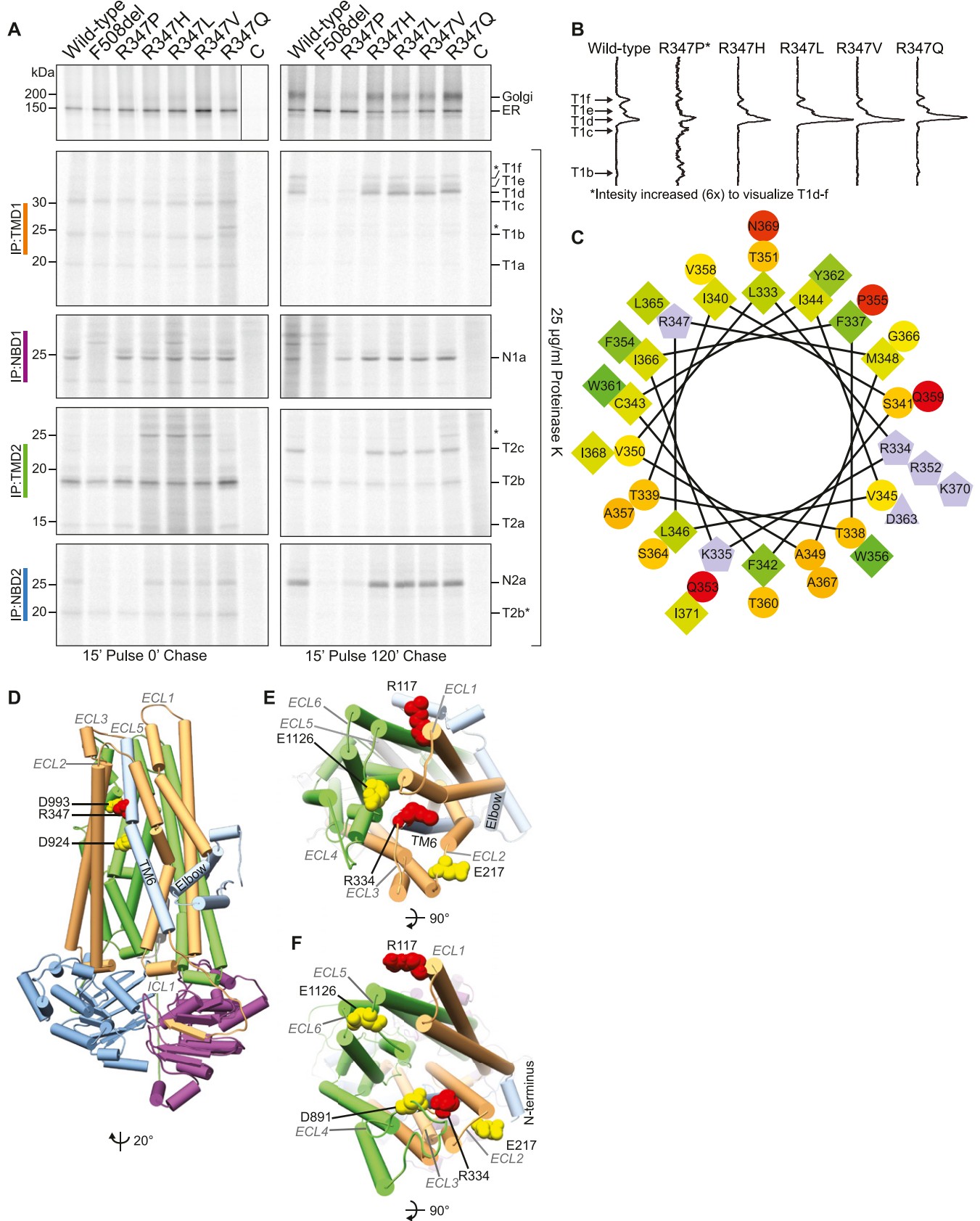

Van Goor et al, 2014). R347V, L, and Q CFTR also folded wild-type–like, but the subtle change in TMD1 as reflected by the enrichment of the T1d-fragment (Fig 5A and B) suggests a subtle conformational defect at the N terminus, consistent with the subtle functional defect of R347H CFTR (Cotten & Welsh, 1999; Van Goor et al, 2014).

Other class-IV mutants fold largely like wild-type CFTR albeit with some minor differences. R334 is in ECL3 with its side chain towards TM1 (Fig 5E). This must be a more tolerant part of the molecule because even introduction of a large aromatic residue such as Trp does not grossly perturb folding of the protein. In the open channel structure model (Mornon et al, 2015), R334 is located close to ECL4 in TMD2, where R334 may have a prominent role in maintaining a stable open form by making a salt bridge with D891 (Sheppard et al, 1993) and in the closed state with E217 (Rahman et al, 2013) (Fig 5E and F).

The lack of differences between wild-type CFTR and the R117 mutants suggested that the gross conformation of the channel is not altered. R117 is in the first extracellular loop (ECL1) of TMD1 at the beginning of TM2 (Fig 5D and F) and creates or stabilizes the open channel through a salt bridge with E1126 (Sheppard et al, 1993; Rahman et al, 2013; Cui et al, 2014; Yu et al, 2016). Lack of large structural effects in both ECL mutants suggests a minor role of ECLs in overall folding but a major role in maintaining an open pore.

An increasing number of human diseases are known to be caused by misfolding of mutant ABC transporters (Theodoulou & Kerr, 2015). Screens for small molecule treatments have been instigated, including for Tangier disease (ABCA1), gout (ABCG2), Stargardt eye disease (ABCA4), progressive familial intrahepatic cholestasis type 2 (ABCB11) (Allikmets et al, 1997; Strautnieks et al, 1998; Rust et al, 1999; Woodward et al, 2009), and congenital hyperinsulinemia (Chen et al, 2013). Unlike the last example where carbamazepine corrects the folding of the sulfonylurea receptor 1, corrector compounds and robust functional readouts are not as readily available for most ABC-transporter diseases as for CFTR. In those cases, an in-depth analysis of the folding–function relationship of the target protein provides an attractive approach for characterization of the mode of action of novel therapeutic compounds as we have shown here for CFTR.

# Materials and Methods

## Reagents

VX-809 and VX-770 (Selleckchem) were dissolved in DMSO and stored at −80°C. Amiloride, forskolin, genistein, carbachol, 4,4′-diisothiocyano-2,2′-stilbenedisulfonic acid (all from Sigma-Aldrich) were dissolved in water and stored at −20°C. Linear 40-kD polymer polyethylenimine (Polysciences) was dissolved in water to 1 mg/ml, pH 7.0; this solution will be referred to as PEI from now on. The rabbit antibody Mr. Pink and mouse monoclonal 596 against CFTR have been characterized before (Gentzsch et al, 2003; Hoelen et al, 2010). The antibodies TMD1C and TMD2C were generated in rabbits against KLH-coupled peptides SLGAINKIQDFLQKQEYK and EGKPTKSTKPYKNGQ, respectively.

Mutations in R347V, R347L, and R347Q were made from pBi.CMV2-CFTR template with site-directed mutagenesis, using primers 5′ ctgcattgttctggtcatggcggtcactc 3′, 5′ ctgcattgttctgctcatggcggtcactc 3′, and 5′ ctgcattgttctgcagatggcggtcactc 3′, respectively.

## Cell culture and transient expression

HEK293 cells were maintained in DMEM supplemented with 10% fetal bovine serum (qualified, E.U.–approved, South America origin) and 2 mM GlutaMAX at 37°C and 5% $CO_2$ unless specified otherwise. All reagents were purchased from Life Technologies. cDNA encoding wild-type or mutant CFTR was introduced in the NotI site of the bicistronic plasmid pBi.CMV2 (Clontech), which also drives expression of GFP. HEK293 cells were grown in 6- or 3.5-cm dishes to ~70% confluency and transfected using PEI. 5 μg of DNA was added to 200 μl 150 mM NaCl and 12.5 μl PEI for 6-cm dishes and 2.5 μg DNA was added to 100 μl 150 mM NaCl and 7.5 μl PEI for 3.5-cm dishes; these solutions were incubated for 10 min at RT. The cells were then grown in DMEM supplemented with 5% FBS and 1 mM GlutaMAX, and DNA:PEI complexes were added. After 4 h, the medium was aspirated and replaced with full growth medium.

## Pulse-chase analysis

HEK293 cells were used 24 h post-transfection for pulse-chase experiments as described (Braakman et al, 1991; Hoelen et al, 2010). Briefly, the cells were labeled for 15 min with 143 μCi/6-cm dish of EasyTag Express $^{35}$S-protein-labeling mix (Perkin Elmer). Label incorporation was stopped by adding an excess unlabeled cysteine and methionine. At indicated chase times, dishes were transferred to ice and washed twice with ice-cold HBSS (Life Technologies). The cells were lysed using 1% Triton X-100 in MNT (20 mM MES, 100 mM NaCl, and 30 mM Tris–HCl, pH 7.5). The lysates were cleared by centrifugation at 16,000 g and 4°C. The supernatant was used for limited proteolysis or immunoprecipitation as described (Jansens et al, 2002). Where indicated, VX-770 or VX-809 was

**Figure 5. The proline in R347P destabilizes TM6.**
**(A)** As in Fig 3B, HEK293 cells transiently expressing indicated CFTR constructs were labeled for 15 min and chased for 0 or 2 h. The cells were lysed in 1% Triton X-100 in MNT and incubated with proteinase K for 15 min, followed by immunoprecipitation of CFTR with domain-specific antibodies. The lane labeled C is a nontransfected control. * indicates nonspecific bands. **(B)** Lane profiles of IP:TMD1 data shown in (A). **(C)** Helical wheel representation of CFTR residues 331–360, showing that R347 is largely surrounded by small hydrophobic residues. Hydrophilic (circle), hydrophobic (diamond), negatively charged (triangle), positively charged (pentagons). Shades of green indicate the hydrophobicity with dark green having high hydrophobicity and yellow zero hydrophobicity. Hydrophilic residues are shades of red, more intense red means most hydrophilic residue. Potentially charged residues are light blue. **(D)** Cryo-EM structure of human CFTR illustrating R347 and interacting residues, helices are shown as cylinders (Liu et al, 2017) (PBD: 5UAK). R347 is highlighted in red, and its interacting residues D993 and D924 are colored yellow. Transmembrane helix 6 and the N terminus (AA 1–77) are colored in light blue. Extracellular loops and intracellular loop 1 (ICL1) are labeled in dark gray. The colors wheat, blue, gray, green, and purple indicate TMD1, NBD1, R region, TMD2, and NBD2, respectively. The arrow indicates rotation in relation to Fig 3A left panel. **(E)** Same as in D, R117 in ECL1 and R334 in ECL2 are shown as red spheres, and their interacting residues E1126 in ECL5 and E217 in ECL3 are represented as yellow spheres. **(F)** Structural model of CFTR in the open conformation (Mornon et al, 2015), illustrating R117 and R334 in red along with their binding partners in yellow. Colors and annotations are the same as in (D, E).
Source data are available online for this figure.

added to 3 $\mu$M (final concentration) medium during starvation, pulse, and chase.

## Limited proteolysis

Limited proteolysis was performed as described (Kleizen et al, 2005; Hoelen et al, 2010). In short, detergent cell lysates were digested for 15 min on ice, using 25 $\mu$g/ml proteinase K (Sigma-Aldrich). The reaction was stopped by adding an equal volume of MNT supplemented with 1% Triton X-100, 2 mM PMSF, and 2 $\mu$g/ml of chymostatin, leupeptin, antipain, and pepstatin A (CLAP; all from Sigma-Aldrich).

## Immunoprecipitation

Following limited proteolysis, the samples were transferred to 50 $\mu$l of protein-A beads (GE Healthcare) that had been preincubated with the antibody for 10 min at 4°C. All immunoprecipitates were washed twice for 10 min at RT. Proteolytic fragments originating from TMD1 were immunoprecipitated with 4 $\mu$l of TMD1C antibody, added to protein-A beads, incubated for 3 h at 4°C, and washed with 10 mM Tris–HCl, pH 8.6, 300 mM NaCl, 0.05% SDS, and 0.05% Triton X-100. NBD1 fragments were immunoprecipitated with 5 $\mu$l of Mr. Pink antibody and incubated overnight at 4°C. The immunoprecipitates were washed with 10 mM Tris–HCl, pH 8.6, 300 mM NaCl, 0.1% SDS, and 0.05% Triton X-100. Fragments derived from TMD2 were immunoprecipitated with 4 $\mu$l of TMD2C antibody for 3 h at 4°C and washed with 50 mM Tris–HCl, pH 8.0, 150 mM NaCl, and 1 mM EDTA. NBD2-derived fragments were immunoprecipitated with 0.25 $\mu$l 596 antibody and incubated for 3 h at 4°C and washed using 30 mM Tris–HCl, pH 7.5, 20 mM MES, 100 mM NaCl, and 0.5% Triton X-100. Washed immunoprecipitates received 10 $\mu$l 10 mM Tris–HCl, pH 6.8, and 10 $\mu$l 2× reducing Laemmli sample buffer and were heated for 5 min at 55°C.

## Cell-surface biotinylation

Four hours post-transfection, the cells received fresh medium containing 3 $\mu$M VX-770, 3 $\mu$M VX-809, or no drug and were subsequently grown for 16 h. The dishes were transferred to ice and washed twice with PBS[++] (137 mM NaCl, 2.7 mM KCl, 8 mM Na$_2$HPO$_4$, 2 mM KH$_2$PO4, 1 mM CaCl$_2$, and 0.5 mM MgCl$_2$) and left to cool for 15 min. Buffer was aspirated, and 500 $\mu$l PBS[++] containing 0.5 mg/ml sulfo-NHS-SS-biotin (Thermo Fisher Scientific) was added and incubated on ice After 30 min; the buffer was removed and remaining nonreacted sulfo-NHS-SS-biotin was quenched by washing twice with 1% BSA in PBS[++] (Sigma-Aldrich). The cells were lysed in 300 $\mu$l MNT containing 1% Triton X-100, 1 mM CLAP, and 1 mM PMSF. The lysates were cleared by centrifugation at 16,000$g$ and 4°C. Directly after centrifugation, 10 $\mu$l of the lysate was transferred to an equal volume of 2× reducing Laemmli sample buffer and saved as input sample. The remaining lysate was incubated with 25 $\mu$l NeutrAvidin beads (Thermo Fisher Scientific) for 1 h. The NeutrAvidin beads were washed twice with 10 mM Tris–HCl, pH 8.6, 300 mM NaCl, 0.1% SDS, and 0.05% Triton X-100 before receiving 15 $\mu$l of 10 mM Tris–HCl, pH 6.8, and 15 $\mu$l of 2× reducing Laemmli sample

buffer. The samples were resolved by SDS–PAGE and analyzed by Western blot.

## Western blot

The samples were run on 7.5% SDS-PAA gels and transferred to PVDF membrane (Millipore) under 38 V for 18-21 h at 4 degrees. After transfer blots were rinsed in PBS (137 mM NaCl, 2.7 mM KCl, 8 mM Na$_2$HPO$_4$, and 2 mM KH$_2$PO$_4$; 1 mm) and ultrapure water, they were subsequently incubated in LiCoR block buffer, diluted 1:1 in PBS (BB), for 1 h at RT. CFTR was detected using 596 (1:5,000) and actin with rabbit anti-actin (1:5,000; Sigma-Aldrich) in BB with 0.1% Tween-20 for 1 h at RT. The blots were washed four times in PBS containing 0.1% Tween-20 before incubation with goat-anti-mouse Alexa-800 (1:10,000; LiCoR) and donkey-anti-rabbit Alexa-680 (1:10,000; LiCoR) in BB supplemented with 0.1% Tween-20 and 0.02% SDS for 1 h at RT. Before imaging, the blots were washed as before with an additional wash step in PBS before imaging on LiCoR Odyssey CLx according to the manufacturer's specifications.

## ICMs

Transepithelial chloride secretion in human rectal biopsies (four per subject) was measured as described (De Jonge et al, 2004), with a prewash modification (De Boeck et al, 2011) to enhance the forskolin-induced anion secretion. Briefly, the biopsies were kept in PBS on ice and mounted in micro-Ussing chambers (aperture 1.13 or 1.77 mm$^2$). After the chambers were equilibrated, the compounds were added to the mucosal (M) or serosal (S) side in the following order: (1) amiloride (0.1 mM, M) to block amiloride-sensitive electrogenic Na+ absorption, (2) forskolin (10 $\mu$M, M+S) plus 3-Isobutyl-1-methylxanthine (100 $\mu$M, M+S) to activate CFTR dependent chloride secretion, (3) VX-770 (20 $\mu$M, M+S) and/or genistein (50 $\mu$M, M+S) for potentiation of CFTR channels, and (5) carbachol (100 $\mu$M, S) to activate basolateral Ca$^{2+}$-dependent K$^+$ channels and increase the electrochemical driving force for CFTR-mediated chloride secretion. Isc values ($\mu$A) were divided by the surface area of the tissue holder to obtain $\mu$A·cm$^{-2}$.

## Crypt isolation and organoid culture from human rectal biopsies

Crypt isolation and culture of human intestinal organoids has been described (Sato et al, 2011; Dekkers et al, 2013). Biopsies obtained for ICMs were washed with 10 mM EDTA in PBS for 30 (small intestine) or 90–120 (rectum) min. at 4°C. The supernatant was discarded and EDTA was washed away. Crypts were isolated by rapid up-and-down pipetting of the biopsies in clean PBS, upon which crypts are released from the tissue. The isolated crypts in PBS were transferred to a clean tube, centrifuged at 130 $g$, and embedded in 50% Matrigel (growth factor reduced, phenol free, BD Bioscience) and seeded (30 $\mu$l Matrigel per well, 50–200 crypts) in 24-well plates. Afterward, Matrigel was solidified by incubation for a minimum of 15 min at 37°C and then DMEM/F12 advanced medium was added, supplemented with Primocin 100 $\mu$g/ml, 2 mM GlutaMax, N2, B27 (all from Invitrogen), 1 $\mu$M N-acetylcysteine (Sigma-Aldrich), and the following growth factors: 50 ng/ml mouse epidermal growth factor, 50% Wnt3a-conditioned medium and 10% Noggin-conditioned

medium, 20% Rspo1-conditioned medium, 10 $\mu$M nicotinamide (Sigma-Aldrich), 10 nM gastrin (Sigma-Aldrich), 500 nM A83-01 (Tocris), and 10 $\mu$M SB202190 (Sigma-Aldrich). The medium was refreshed every 2–3 d, and organoids were passaged every 7–10 d.

### FIS assay

Organoids from 7-d-old cultures were plated in 4 $\mu$l of 50% Matrigel into a 96-well flat bottom plate (Nunc) immersed in 100 $\mu$l of culture medium as described earlier. 1 d after seeding, the organoids were stained with 10 $\mu$M calcein green for 30 min. Forskolin was added at indicated concentrations, and the swelling was monitored for 1 h; images were captured every 10 min, using a live-cell confocal microscope (LSM710, Zeiss, 5× magnification). VX-809 (3 $\mu$M) was preincubated 20–24 h before measurement, whereas VX-770 (3 $\mu$M) was added directly. FIS was quantified using Volocity imaging software (Improvision). Organoid surface area (xy plane) at t = 60 min was calculated relative to that of the surface area at t = 0 and averaged over three biological replicates per condition. The AUC was calculated using Prism (GraphPad). Graphs were produced in Tibco Spotfire (Perkin Elmer).

### Other methods

CFTR structure from (Liu et al, 2017) PDB: 5UAK and structural model from (Mornon et al, 2015). Images of protein structures were generated using Chimera software (Pettersen et al, 2004), and helical wheel representations were made using http://www.tcdb.org/progs/helical_wheel.php. Western blot images were processed using Image Studio 5.2 (LiCoR).

### Human material

Approval for this study was obtained by the ethics committees of the University Medical Center Utrecht and Erasmus MC Rotterdam. Consent was obtained from all subjects participating in the present study. Rectal organoids were generated from biopsies after ICMs were obtained during standard cystic fibrosis care.

## Supplementary Information

## Acknowledgements

We thank Dr. Phil Thomas and Linda Millen (Department of Physiology, UT South Western Medical Center, Dallas) for generously donating the CFTR mutants, Dr. Inez Bronsveld (Department of Pulmonary Diseases University Medical Center Utrecht, Utrecht) for providing the ICM equipment, and Dr. John Riordan, University of North Carolina—Chapel Hill, Cystic Fibrosis Foundation Therapeutics, for providing antibodies. We thank Dr. Paola Vergani for critically reading the manuscript and Drs. Paola Vergani and David Sheppard for sharing unpublished results. This work was funded by the Netherlands Organization for Health Research and Development (Zon-Mw TOP grant 40-00812-98-14103 to CK van der Ent, JM Beekman, I Braakman), the Netherlands Cystic Fibrosis Foundation (HIT-CF grant), the Cystic Fibrosis Foundation (BRAAKM14XX0 to I Braakman), and Vertex Pharmaceuticals, Gilead, and Galapagos, including fees for serving on an advisory board paid to his institution from ProQR, and lecture fees paid to his institution from Teva Pharmaceuticals (CK van der Ent).

## Author Contributions

M van Willigen: conceptualization, data curation, formal analysis, validation, investigation, visualization, methodology, and writing—original draft, review, and editing.
AM Vonk: formal analysis, validation, investigation, methodology, and writing—review and editing.
HY Yeoh: resources, investigation, and methodology.
E Kruisselbrink: formal analysis, methodology, and writing—review and editing.
B Kleizen: formal analysis and investigation.
CK van der Ent: conceptualization, formal analysis, supervision, funding acquisition, investigation, and writing—review and editing.
MR Egmond: formal analysis and writing—review and editing.
HR de Jonge: conceptualization, formal analysis, supervision, funding acquisition, validation, investigation, visualization, methodology, and writing—review and editing.
I Braakman: conceptualization, resources, formal analysis, supervision, funding acquisition, investigation, methodology, and writing—original draft, review, and editing.
JM Beekman: conceptualization, formal analysis, supervision, funding acquisition, investigation, methodology, and writing—review and editing.
P van der Sluijs: conceptualization, formal analysis, supervision, investigation, visualization, and writing—original draft, review, and editing.

### Conflict of Interest Statement

CK van der Ent received funding from Vertex Pharmaceuticals.

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
