## [Reviewer comments · Life Science Alliance]

Life Science Alliance

Folding-function relationship of the most common Cystic-Fibrosis-causing conductance mutants

Marcel van Willigen, Annelotte Vonk, Hui Ying Yeoh, Evelien Kruisselbrink, Bertrand Kleizen, Cornelis van der Ent, Maarten Egmond, Hugo de Jonge, Ineke Braakman, Jeffrey Beekman, and Peter van der Sluijs

DOI: [10.26508/lsa.201800172](https://doi.org/10.26508/lsa.201800172)

Corresponding author(s): Peter van der Sluijs, Utrecht University and Ineke Braakman, Utrecht University

Review Timeline:

Submission Date:	2018-08-24
Editorial Decision:	2018-09-18
Revision Received:	2018-12-30
Editorial Decision:	2019-01-08
Revision Received:	2019-01-09
Accepted:	2019-01-10

Scientific Editor: Andrea Leibfried

Transaction Report:

September 18, 2018

Re: Life Science Alliance manuscript #LSA-2018-00172-T

Dr. Peter van der Sluijs
Utrecht University
CPC-Chemistry
Padualaan 8
Utrecht 3584CH
NETHERLANDS

Dear Dr. van der Sluijs,

Thank you for submitting your manuscript entitled "Folding-function relationship of the most common Cystic-Fibrosis-causing conductance mutants" to Life Science Alliance. The manuscript was assessed by expert reviewers, whose comments are appended to this letter.

As you will see, the reviewers think that your work nicely confirms that CFTR R347P is not a typical class IV but rather a class II mutant as previously proposed, and they appreciate the systematic analysis performed. They think, however, that the work needs to be revised to allow publication in Life Science Alliance.

Importantly, some re-writing is needed (reviewer #1 and #2), organoids from homozygous individuals should get included if the material is available (reviewer #2), the sample size needs to get increased (reviewer #2), and chloride conductance assays should be added (reviewer #3). I don't mention the other issues noted by the reviewers here, as they all seem straightforward to address.

Thank you for this interesting contribution to Life Science Alliance. We are looking forward to

receiving your revised manuscript.

Sincerely,

- A letter addressing the reviewers' comments point by point.
- An editable version of the final text (.DOC or .DOCX) is needed for copyediting (no PDFs).
- High-resolution figure, supplementary figure and video files uploaded as individual files: See our detailed guidelines for preparing your production-ready images, <http://life-science-alliance.org/authorguide>
- Summary blurb (enter in submission system): A short text summarizing in a single sentence the study (max. 200 characters including spaces). This text is used in conjunction with the titles of papers, hence should be informative and complementary to the title and running title. It should describe the context and significance of the findings for a general readership; it should be written in the present tense and refer to the work in the third person. Author names should not be mentioned.

B. MANUSCRIPT ORGANIZATION AND FORMATTING:

Full guidelines are available on our Instructions for Authors page, <http://life-science-alliance.org/authorguide>

Reviewer #1 (Comments to the Authors (Required)):

Van Willigen et al. assessed several class-IV mutations in CFTR based on their structure and function to see how they respond to correctors and potentiators. The mutations the authors used were R117C/H, R334W, and R347P. Out of these R347P, is proven to be the least efficient in maturing. These mutations were initially class IV but have been re-classified as mixed class III/IV. The authors used several techniques such as Forskolin Swelling Assays in organoids to measure CFTR function, ICM (intestinal current measurements) to assess the functionality of mutated CFTR channels, and radiolabeling coupled to a protease-sensitivity assay to determine CFTR function and structure. A principal conclusion from the study was that among those mutations, R347P shows a lack of function and a misfolded conformation, more closely resembling a class II mutation.

This manuscript does increase our understanding as to how best to deploy CFTR potentiators and correctors to evaluate various CFTR CF-phenotypic mutants in a therapeutic setting. Nevertheless, it seems to this reviewer to be heavily overwritten, with many lengthy sentences and paragraphs giving in-depth suggestions as though there is some unspoken need to explain why every given mutant behaved a certain way. Among numerous examples, at the top of p. 20, the authors write that "R117H is less defective and easier to correct than R117C...probably because the positively charged histidine does substitute well for R117's role in maintaining the pore". This is a 'tautological' sentence, and does not add to the discussion. In this aspect, the paper needs to be extensively shortened and streamlined. Perhaps the strength of the paper lies in the multi-faceted use of techniques that can generally be systematically employed to measure corrector/potentiator response(s) to any given CFTR mutant.

Specific questions

- Where is the N2b band that they refer to on page 12, and have the authors labeled N2a correctly? (i.e., was that supposed to be N2b instead?)
- Page 13: The authors refer several times to the results for deltaF-508 but these results are not referenced? Published in a previous study?
- Page 15: The authors mention that VX-809 made R347P functionally responsive to VX-770 but they did not see this result in their biochemical assay. What could be the explanation/hypothesis for this?

Typos:

- Page 3 (abstract): "criterium" should be changed to criterion
- Page 18: second last line: "does causes" should be changed to just "causes"
- Page 28: Section titled Problem: "does" should be changed to "do" in the 10th line
- Page 28: Section titled Results: second line criterium should be changed to criterion.

Reviewer #2 (Comments to the Authors (Required)):

The authors use existing functional assays of Fsk-induced organoid swelling and Ussing chamber studies, along with pulse-chase and limited proteolysis experiments to suggest from F508del/R347P and other class IV mutation heterozygous patient rectal organoid samples that the R347P mutation is misclassified and should be considered a class II mutation, not showing function unless corrected with VX-809, while R117H and R334W do yield functional responses in the functional assay and complex glycosylated protein. The authors suggest that insertion of a proline at the 347 residue rather than specifically the lack of the arginine residue as the cause of the

defect. While not highly innovative in nature, the study does provide useful insight about the R347P defect and potential clinical avenues that will be required to treat it.

Major Comments:

The major drawback in the study is that heterozygotes with F508del are used to study all of the mutations in question. Homozygous individuals, while more difficult to obtain, would very advantageous here. The authors take steps to mitigate this drawback by studying the mutations in the absence of VX-809 corrector where F508del is typically acknowledged to have little effect. The authors also describe a study of a stop mutation in a heterozygous sample and demonstrate low function in these samples even in the presence of both corrector and potentiator treatments (pg 9 and figure 2). The authors make the argument that minimal function in this heterozygote indicates that the F508del allele will have little contribution in any of the other heterozygotes studied. Several studies (e.g. see for example Pranke et al 2017), have suggested that there may be considerable variability in F508del homozygote function and response to modulator compounds, in a variety of assays. What evidence supports or refutes this variability in F508del signal in the system described here? How can we be sure that the F508del allele contributes nothing in all heterozygote sample function described here?

How many individuals with each genotype are studied? It appears to just be one individual for each mutation of interest. Again, there are many factors that can influence an observed functional response in an assay, including complex alleles (other mutations in CFTR), modifier genes, etc. in an individual patient sample. It would be helpful to confirm similar effects in multiple patient samples rather than one single patient. Figure 2 has no error bars. How many samples were averaged for these data? At minimum there should be several replicates on the same individual if there aren't multiple individual's samples available for the study. How do you determine significance if there are no error bars? Related to this, how do you determine the level of improvement (i.e. minimal improvement)? There seems to be shifts in the EC50s for forskolin which is interesting. Can you compare that and what does that mean?

Page 8/Figure 1: The authors mentioned that VX-770 doesn't potentiate CFTR function (page 8 and figure legend 1). The authors have shown traces with Genistein but not VX-770. The authors should show typical traces with VX-770 treatment. Authors should show CFTR inhibitor treatment in the traces to confirm that CFTR-specific current traces are being presented. It would be beneficial to show results for F508del/E730X and R334W/R746W patients. F508del/E730X is a priority as the rationale for the F508del allele effect should be accounted for as you have mentioned in the Results for Figure 2, especially when you have indicated that "ICM (Figure 1) and FIS (Figure 2) strongly correlate but each patient-derived organoid is unique and each patient may respond differently to drug treatment." (page 9) Is the function related to amount of protein expressed at the cell surface? Should show a western beside each patient sample to support this. Figure 3 to 5: There appears to be different amounts of total protein amongst the different mutants. How do you quantify the efficiency of transport from the ER when there are different protein amounts? Perhaps show a quantification (with LICOR or ImageJ for densitometry analysis). There are more bands present than the ones you have indicated on the immunoblots. What are these bands? Have you tried different antibodies on the domains and do the same bands light up with different antibodies? Main issue with these figures: the exposures of the blots for all domains are very different across the figures. Please keep them at similar exposures so that it would make it easier for readers to compare the figures.

Minor comments:

Figure 1: Results section indicated that forskolin and IBMX were used but the traces only show forskolin addition.

Figure 2: It would be better to label each graph with a letter (i.e. A-C) and use the same in the manuscript. In figure 2, the authors compare the R347P/F508del response after ORKAMBI treatment with F508del/E730X response. It would be helpful to compare the ORKAMBI response in intestinal organoids from a homozygous F508del CF patient. Where is the wildtype healthy control for this?

Why did you use 3 μ M VX-770? Have you used different concentrations (i.e. lower concentrations) and does it still have that destabilizing effect that you observed?

Discussion as to why R347P has such a profound effect on exit from ER to Golgi (trafficking) compared to R347V and R347L (which are all changes from arginine to hydrophobic amino acid that should also disrupt the helix) was not very convincing. The "subtle change in TMD1 suggests a conformational defect in the N-terminus"? How? Some of the peaks were higher than Wt? (Figure 5B) This part is unclear in the Discussion.

Page 14, line 11: These results are in agreement with recently publication by Laselva O et al. (2018), where the authors showed that VX-809 corrects F508del-CFTR by interacting the two membrane spanning domains. The authors may want to cite this finding.

Page 15, line 10: are the authors referring to Figure 5B or Figure 4B?

Proofread the manuscript carefully: e.g. pg 6 "Above mutants", pg 18 "therefore does causes"

Reviewer #3 (Comments to the Authors (Required)):

The authors here have addressed the molecular basis of the channel dysfunction associated with the Class IV mutants of CFTR and especially R347P. It is an important and well-executed study and I highly recommend its publication.

In my opinion the present study and its presentation should be strengthened by addressing these two issues:

1. While characterizing the effect of proline substitution at position 347, the authors have studied the effect of various amino acid substitutions on the folding property of the protein. A parallel analysis of how this mutants fare in the cell by chloride conductance assays would show whether the conclusions drawn by the authors are tenable in vivo.
2. The authors clearly show that VX-809 protects the R347P mutant. Studies on the mechanism of action of VX-809 have showed how it may protect DF508 mutant. A discussion of these studies and how it compares with the authors' own explanation of the mode of action of VX-809 on R347P mutant is needed to provide a comprehensive view of our understanding.

Utrecht, December 30, 2018

Reviewer #1:

Van Willigen et al. assessed several class-IV mutations in CFTR based on their structure and function to see how they respond to correctors and potentiators. The mutations the authors used were R117C/H, R334W, and R347P. Out of these R347P, is proven to be the least efficient in maturing. These mutations were initially class IV but have been re-classified as mixed class II/III/IV. The authors used several techniques such as Forskolin Swelling Assays in organoids to measure CFTR function, ICM (intestinal current measurements) to assess the functionality of mutated CFTR channels, and radiolabeling coupled to a protease-sensitivity assay to determine CFTR function and structure. A principal conclusion from the study was that among those mutations, R347P shows a lack of function and a misfolded conformation, more closely resembling a class II mutation.

This manuscript does increase our understanding as to how best to deploy CFTR potentiators and correctors to evaluate various CFTR CF-phenotypic mutants in a therapeutic setting. Nevertheless, it seems to this reviewer to be heavily overwritten, with many lengthy sentences and paragraphs giving in-depth suggestions as though there is some unspoken need to explain why every given mutant behaved a certain way. Among numerous examples, at the top of p. 20, the authors write that "R117H is less defective and easier to correct than

R117C....probably because the positively charged histidine does substitute well for R117's role in maintaining the pore". This is a 'tautological' sentence, and does not add to the discussion. In this aspect, the paper needs to be extensively shortened and streamlined. Perhaps the strength of the paper lies in the multi-faceted use of techniques that can generally be systematically employed to measure corrector/potentiator response(s) to any given CFTR mutant.

We have removed the tautological text on the previous page 20 and went through the entire text to make it more succinct. We hope this version is more to the liking of reviewer.

Specific questions

- Where is the N2b band that they refer to on page 12, and have the authors labeled N2a correctly? (i.e., was that supposed to be N2b instead?)

We had mislabeled the fragment and corrected this in Figure 3B. We have identified the 2 bands immunoprecipitated with the NBD2 antibody as a proper fragment of NBD2, N2a, and a C-terminal TMD2 fragment, T2b*, which is slightly larger than T2b and almost invisible above T2b in the TMD2 immunoprecipitation.

- Page 13: The authors refer several times to the results for deltaF508 but these results are not referenced? Published in a previous study?

This description referred to data in Figure 3B (and can be seen as well in Figure panels 4A and 5A), which were described on the previous page 11; we have added clarification from page 11, bottom 4 lines, to page 13, first paragraph.

- Page 15: The authors mention that VX-809 made R347P functionally responsive to VX-770 but they did not see this result in their biochemical assay. What could be the explanation/hypothesis for this?

Inspired by this reviewers' and reviewer-2's questions (see also below) we went through our published organoid data and have used the average data from eight F508del/F508del organoids (Figure 3A in Dekkers et al, Sci Transl Med, 2016) to correct the signals of the R347P/F508del organoid in Figure 2A. We subtracted 50% of the average response of these eight F508del/F508del organoids to yield an estimation of the 'net' signal of R347P in the organoid-swelling assay as now shown in Figure 2A'. We did the same for the other organoid tracings in Fig. 2B' and C' and added clarifying text on page 9, from line 13. Figure 2A' now shows a lack of response of R347P CFTR to VX-770, in both absence and presence of VX-809, consistent with our biochemical data. We thank both reviewers 1 and 2 for their insightful comments and apologize for this oversight.

Typos:

-Page 3 (abstract): "criterium" should be changed to criterion

-Page 18: second last line: "does causes" should be changed to just "causes"

-Page 28: Section titled Problem: "does" should be changed to "do" in the 10th line

-Page 28: Section titled Results: second line criterium should be changed to criterion.

We corrected the typos at the corresponding and other places

Reviewer #2 :

The authors use existing functional assays of Fsk-induced organoid swelling and Ussing chamber studies, along with pulse-chase and limited proteolysis experiments to suggest

from F508del/R347P and other class IV mutation heterozygous patient rectal organoid samples that the R347P mutation is misclassified and should be considered a class II mutation, not showing function unless corrected with VX-809, while R117H and R334W do yield functional responses in the functional assay and complex glycosylated protein. The authors suggest that insertion of a proline at the 347 residue rather than specifically the lack of the arginine residue as the cause of the defect. While not highly innovative in nature, the study does provide useful insight about the R347P defect and potential clinical avenues that will be required to treat it.

Major Comments:

The major drawback in the study is that heterozygotes with F508del are used to study all of the mutations in question. Homozygous individuals, while more difficult to obtain, would be very advantageous here. The authors take steps to mitigate this drawback by studying the mutations in the absence of VX-809 corrector where F508del is typically acknowledged to have little effect.

The authors also describe a study of a stop mutation in a heterozygous sample and demonstrate low function in these samples even in the presence of both corrector and potentiator treatments (pg 9 and figure 2). The authors make the argument that minimal function in this heterozygote indicates that the F508del allele will have little contribution in any of the other heterozygotes studied. Several studies (e.g. see for example Pranke et al 2017), have suggested that there may be considerable variability in F508del homozygote function and response to modulator compounds, in a variety of assays. What evidence supports or refutes this variability in F508del signal in the system described here? How can we be sure that the F508del allele contributes nothing in all heterozygote sample function described here?

We thank reviewer for these insightful comments: the functional measurements in organoids indeed integrate both CFTR alleles as well as other modifiers that act on CFTR ion transport (such as non-CFTR ion channels/transporters that impact the electro-chemical gradient). To directly deduce patient-specific single-allele function would require allele-specific genetic manipulation of the organoid genome. We indeed cannot rule out that individual allele-specific F508del function may contribute to an observation.

To address this issue (which was raised by reviewer 1 as well, see above) we had published the average F508del/class I response of eight individuals (Fig. 3A in Dekkers et al, Sci Transl Med, 2016). By subtracting this average response from the corresponding individual curves (DMSO, VX-770, VX-809, or VX-770+VX-809), we had estimated the impact of the other allele. In this way, we deduced that N1303K likely is not associated with function, whereas A455E is, with a differential response to modulator (Dekkers et al, Eur. Resp. J. 2016).

Similarly, we have corrected the signals of R347P/F508del in Figure 2A by subtracting 50% of the response of eight F508del/F508del organoids (Figure 3A in Dekkers et al, Sci Transl Med, 2016) to yield an estimation of the 'net' signal of R347P in the organoid swelling as shown in Figure 2A'. This showed that R347P is associated with minimal function, which now is more in line with the results of the folding experiments. We added clarifying text on page 9, from

line 13. We also applied this correction for the R117H/F508del and R334W/mixed allele response curves to generate Figure 2B' and Figure 2C', respectively.

How many individuals with each genotype are studied? It appears to just be one individual for each mutation of interest. Again, there are many factors that can influence an observed functional response in an assay, including complex alleles (other mutations in CFTR), modifier genes, etc. in an individual patient sample. It would be helpful to confirm similar effects in multiple patient samples rather than one single patient. Figure 2 has no error bars. How many samples were averaged for these data? At minimum there should be several replicates on the same individual if there aren't multiple individual's samples available for the study.

We performed measurements with additional patient organoids and now have for each genotype triplicate measurements in organoids of 3 individuals (Figure 2A-C'), as indicated in the legend of Figure 2.

How do you determine significance if there are no error bars? Related to this, how do you determine the level of improvement (i.e. minimal improvement)? There seems to be shifts in the EC50s for forskolin which is interesting. Can you compare that and what does that mean? Calculate EC50 values from curves.

We now have statistical data for all curves in Figure 2, which allows reliable conclusions on differences between the organoids. Shifts to the left imply increased sensitivity to forskolin, usually associated with increased function. EC50 values would be useful but are not reliable parameters in this assay, because of 2 reasons:

- CFTR mutants with low functionality respond poorly to forskolin, which does not allow identification of the maximum swelling as it does not plateau, precluding calculation of the EC50; Figure 2A is an example.
- more functional CFTR mutants and wild-type CFTR easily saturate the assay, and plateau at ~3,200 AUC units, because further swelling is not possible. The curve appears to shift but if swelling would not saturate, the EC50 may turn out to be unchanged.

We therefore prefer not to use EC50 values, and added an explanation in the text on page 9, from line 16.

Page 8/Figure 1: The authors mentioned that VX-770 doesn't potentiate CFTR function (page 8 and figure legend 1). The authors have shown traces with Genistein but not VX-770. The authors should show typical traces with VX-770 treatment.

We apologize for this shortcoming, which arose during preparation of the figure, and now have included this information in Figure 1A.

Authors should show CFTR inhibitor treatment in the traces to confirm that CFTR-specific current traces are being presented. It would be beneficial to show results for F508del/E730X and R334W/R746W patients.

In additional experiments, we have recorded intestinal currents in human rectal biopsies of F508del/E730X (Figure 1D) and R334W/R746X patients (assuming that the R746W from this reviewer was a typo and R746X was intended) (Figure 1C).

Unfortunately, CFTR inhibitors (inh-172 and GlyH-101) are poorly soluble and therefore inefficiently targeted to CFTR in the crypts of mucus-rich human rectal biopsies. This

severely hampers the unequivocal dissection of CFTR-mediated vs. non-CFTR-mediated currents in biopsies. According to our experience (shared by other investigators including Alan Verkman, who identified CFTRinh-172), inhibition of CFTR currents by this inhibitor is highly efficient in intestinal cell models such as monolayers of T84 cells or 2D-rectal organoids, but highly variable and only partially effective in human rectal biopsies and in freshly excised mouse intestine. Direct comparison of rectal biopsies from various CF-patients (as in Figure 1A; see also De Winter K et al. Eur. Respir. J. 2018 Sep 17;52(3)) and from healthy controls (Figure 1C) shows unambiguously that the current is virtually completely CFTR-dependent and lost in severe-CF biopsies. To fully rule out the remote possibility that (part of) the current is not carried by CFTR itself but by a (nevertheless completely CFTR-dependent!) other apical chloride channel is a challenge, and also highly unlikely considering the absence of such a channel in organoids generated from colon biopsies.

F508del/E730X is a priority as the rationale for the F508del allele effect should be accounted for as you have mentioned in the Results for Figure 2, especially when you have indicated that "ICM (Figure 1) and FIS (Figure 2) strongly correlate but each patient-derived organoid is unique and each patient may respond differently to drug treatment." (page 9).

We have added data on F508del/E730X, and have used our published average data on F508del/F508del (as explained above for both reviewers 1 and 2) to account for the contribution of the F508del allele. Our recent publication (De Winter-De Groot et al, Eur. Respir. J. 17:52(3), 2018) reports on the correlation between organoid swelling (FIS) and ICM assays, with correlation coefficient r of 0.70. The uniqueness of each patient should be considered relative to the dominant nature of the CFTR genotype: although F508del homozygotes may vary ~3-fold in their response to modulators, none of them have ever reached the response of an R117H-patient organoid.

Is the function related to amount of protein expressed at the cell surface? Should show a western beside each patient sample to support this.

We ascribe to the importance of establishing whether the amount of CFTR at the cell surface is related to function. Assaying the CFTR pool on the apical surface of organoids is technically not feasible due to the lack of access to the lumen of organoids for experimental manipulation. It therefore is not possible to selectively mark the CFTR molecules on the apical surface of organoids. Even if possible, low cell-surface expression would hamper analysis. We nevertheless approached this question of the reviewer with additional cell-surface biotinylation experiments in HEK293 cells transfected with CFTR mutants. The new data have been added as Figure 4B and clearly show that R117C, R117H, and R334W are expressed like wild-type CFTR on the cell surface. R347P however can only be found on the plasma membrane in the presence of VX-809, but not of VX-770. These data are in very good agreement with the biochemical data in Figures 3B and 4A.

Figure 3 to 5: There appears to be different amounts of total protein amongst the different mutants. How do you quantify the efficiency of transport from the ER when there are different protein amounts? Perhaps show a quantification (with LICOR or ImageJ for densitometry analysis).

Because the CFTR variants do not have the same turnover rates, steady-state expression levels do vary. This however will not influence efficiency of transport from ER to the Golgi

complex as we will outline here. The data in the original Figures 3-5 (and present Figures 3B, 4A,C, and 5A) are derived from cells that are metabolically labeled with ³⁵S-methionine/cysteine and in which CFTR is immunoprecipitated with (domain)-specific antibody. The immunoprecipitates were resolved by SDS-PAGE and the ER and Golgi bands were quantitated by phosphor-imaging as detailed in the Methods section. The fraction of CFTR in the Golgi complex then is calculated as intensity of the complex-glycosylated band ('Golgi band') divided by the sum of the Golgi and ER bands, which is independent of expression level. The 0-h chase samples in Figures 3-5 show that the amount of each mutant synthesized and radiolabeled in 15 min of ³⁵S incorporation is very similar, underscoring the similar quantities of CFTR protein the ER needs to handle. Differences arise upon triaging, when wild-type protein and some mutants are released from the ER to travel to the Golgi complex, whereas others are degraded. These differences are apparent in both 2-h chase samples and the wealth of published Western blots of CFTR mutants.

There are more bands present than the ones you have indicated on the immunoblots. What are these bands? Have you tried different antibodies on the domains and do the same bands light up with different antibodies? Main issue with these figures: the exposures of the blots for all domains are very different across the figures. Please keep them at similar exposures so that it would make it easier for readers to compare the figures.

The figure panels with multiple bands derive from lysates of ³⁵S methionine/cysteine-containing proteins that were subjected to limited proteolysis in which peptide fragments of different sizes are generated by proteinase K. These protease-treated lysates were subjected to immunoprecipitation with antibodies specific for TMD1, NBD1, TMD2, and NBD2. Since antibodies to the different domains will have different affinities towards their antigens, signals obtained by them will have different magnitudes.

Minor comments:

Figure 1: Results section indicated that forskolin and IBMX were used but the traces only show forskolin addition.

We have corrected this omission in the revised version of Figure 1.

Figure 2: It would be better to label each graph with a letter (i.e. A-C) and use the same in the manuscript. In figure 2, the authors compare the R347P/F508del response after ORKAMBI treatment with F508del/E730X response. It would be helpful to compare the ORKAMBI response in intestinal organoids from a homozygous F508del CF patient. Where is the wildtype healthy control for this?

We have followed the suggestion of the reviewer and labeled each graph with a letter. As data from homozygous F508del CF patients and healthy controls have been published several times, we added the healthy control to Figure 1, and added text on pages 9 and 10 to relate the class-IV mutants to wild-type and F508del.

Why did you use 3 μM VX-770? Have you used different concentrations (i.e. lower concentrations) and does it still have that destabilizing effect that you observed?

We have titrated VX-770 in our assays and found that a concentration of 0.03 μM already destabilized the N-terminus somewhat and that this increased at higher concentrations (manuscript in preparation); this concentration dependence correlates with the concentration dependence of channel activation. Between 3 and 10 μM VX-770 are typical

concentrations used in the field (Veit et al. *Sci Transl Med.* 2014 Jul 23;6(246) and Cholon et al. *Sci Transl Med.* 2014 Jul 23;6(246)), which were reported to destabilize VX-809-rescued F508del-CFTR in lab cell lines. This effect is visible in our figures for wild-type CFTR at 3 μ M VX-770 (with less Golgi form of CFTR (and therefore less total CFTR) after 2 hours of chase), and is much stronger at 10 μ M. We have chosen 3 μ M to minimize the destabilizing effect on full-length wild-type CFTR.

Discussion as to why R347P has such a profound effect on exit from ER to Golgi (trafficking) compared to R347V and R347L (which are all changes from arginine to hydrophobic amino acid that should also disrupt the helix) was not very convincing. The "subtle change in TMD1 suggests a conformational defect in the N-terminus"? How? Some of the peaks were higher than Wt? (Figure 5B) This part is unclear in the Discussion.

We have addressed this in the Results page 14, lines 4-9, and in the Discussion on page 19, from line 1.

Page 14, line 11: These results are in agreement with recently publication by Laselva O et al. (2018), where the authors showed that VX-809 corrects F508del-CFTR by interacting the two membrane spanning domains. The authors may want to cite this finding.

We now cited the Laselva paper in the discussion section on page 18, line 10.

Page 15, line 10: are the authors referring to Figure 5B or Figure 4B?

The reviewer is correct, we apologize for this omission and corrected the text on page 15, bottom line to page 16, line 4.

Proofread the manuscript carefully: e.g. pg 6 "Above mutants", pg 18 "therefore does

We thank the reviewer for noticing these typos and apologize for the oversight. We carefully checked text and removed these as well as other textual errors.

Reviewer #3 (Comments to the Authors (Required)):

The authors here have addressed the molecular basis of the channel dysfunction associated with the Class IV mutants of CFTR and especially R347P. It is an important and well-executed study and I highly recommend its publication.

In my opinion the present study and its presentation should be strengthened by addressing these two issues:

1. While characterizing the effect of proline substitution at position 347, the authors have studied the effect of various amino acid substitutions on the folding property of the protein. A parallel analysis of how this mutants fare in the cell by chloride conductance assays would show whether the conclusions drawn by the authors are tenable in vivo.

Previous Ussing-chamber-electrophysiology experiments of van Goor et al (*J. Cyst. Fibros.* 13: 29-36, 2014) on cells expressing (amongst others) R347P, R347H, or F508del showed baseline chloride conductance for R347P and F508del and minimal chloride conductance for R347H (which represents the other substitutions: L, V, and Q). These authors also reported that addition of VX-770 improved chloride conductance for R347H but not for the

other patient mutants (including F508del and R347P). Our results of the folding assays with the R347P, R347H mutants with and without VX-770 are in very good agreement with the functional studies of van Goor et al. We have added text on page 19, first paragraph.

2. The authors clearly show that VX-809 protects the R347P mutant. Studies on the mechanism of action of VX-809 have showed how it may protect DF508 mutant. A discussion of these studies and how it compares with the authors' own explanation of the mode of action of VX-809 on R347P mutant is needed to provide a comprehensive view of our understanding.

We have added text on this in the Discussion on page 17, from bottom line.

January 8, 2019

RE: Life Science Alliance Manuscript #LSA-2018-00172-TR

Dr. Peter van der Sluijs
Utrecht University
CPC-Chemistry
Padualaan 8
Utrecht 3584CH
Netherlands

Dear Dr. van der Sluijs,

Thank you for submitting your revised manuscript entitled "Folding-function relationship of the most common Cystic-Fibrosis-causing conductance mutants". As you will see, all reviewers appreciate the introduced changes and we would thus be happy to publish your paper in Life Science Alliance.

Before sending you the official acceptance letter, please log in to your account and fill in the electronic license to publish form. The submission system will now name your manuscript LSA-2018-00172-TRR, please make sure to move all files to this new manuscript version (single click required only). Please also link your profile in our submission system to your ORCID iD, you should have received a message and how to do so.

A. FINAL FILES:

-- High-resolution figure, supplementary figure and video files uploaded as individual files: See our detailed guidelines for preparing your production-ready images, <http://life-science-alliance.org/authorguide>

B. MANUSCRIPT ORGANIZATION AND FORMATTING:

Full guidelines are available on our Instructions for Authors page, <http://life-science-alliance.org/authorguide>

Sincerely,

Reviewer #2 (Comments to the Authors (Required)):

In the revised version of this paper, the authors have responded satisfactorily to the comments/concerns in my original review. Therefore, I recommend that the paper be accepted in its present form for publication in Life Science Alliance, subject to any further revisions that may be recommended by the other reviewers.

Reviewer #3 (Comments to the Authors (Required)):

The authors have adequately addressed all reviewer concerns to the extent possible, have completed multiple additional experiments and have significantly improved the manuscript. The data are strongly supportive of the conclusions. No further issues need to be addressed. Publication of this manuscript is now recommended.

Reviewer #4 (Comments to the Authors (Required)):

I am satisfied with the authors' response to suggestions. I recommend publication with no further changes.

January 10, 2019

RE: Life Science Alliance Manuscript #LSA-2018-00172-TRR

Dr. Peter van der Sluijs
Utrecht University
CPC-Chemistry
Padualaan 8
Utrecht 3584CH
Netherlands

Dear Dr. van der Sluijs,

Thank you for submitting your Research Article entitled "Folding-function relationship of the most common Cystic-Fibrosis-causing conductance mutants". It is a pleasure to let you know that your manuscript is now accepted for publication in Life Science Alliance. Congratulations on this interesting work.

DISTRIBUTION OF MATERIALS:

Again, congratulations on a very nice paper. I hope you found the review process to be constructive and are pleased with how the manuscript was handled editorially. We look forward to future exciting submissions from your lab.

Sincerely,
